# Physiologically based pharmacokinetic/pharmacodynamic model for the prediction of morphine brain disposition and analgesia in adults and children

**Laurens F. M. Verscheijden**[1], **Carlijn H. C. Litjens**[1,2], **Jan B. Koenderink**[1], **Ron H. J. Mathijssen**[3], **Marcel M. Verbeek**[4], **Saskia N. de Wildt**[1,5], **Frans G. M. Russel**[1] *

1 Department of Pharmacology and Toxicology, Radboud Institute for Molecular Life Sciences, Radboud university medical center, Nijmegen, The Netherlands, 2 Department of Pharmacy, Radboud Institute for Health Sciences, Radboud university medical center, Nijmegen, The Netherlands, 3 Department of Medical Oncology, Erasmus MC Cancer Institute, Rotterdam, The Netherlands, 4 Departments of Neurology and Laboratory Medicine, Donders Institute for Brain, Cognition and Behaviour, Radboud university medical center, Nijmegen, The Netherlands, 5 Intensive Care and Department of Paediatric Surgery, Erasmus MC-Sophia Children's Hospital, Rotterdam, The Netherlands

* Frans.Russel@radboudumc.nl

## Abstract

Morphine is a widely used opioid analgesic, which shows large differences in clinical response in children, even when aiming for equivalent plasma drug concentrations. Age-dependent brain disposition of morphine could contribute to this variability, as developmental increase in blood-brain barrier (BBB) P-glycoprotein (Pgp) expression has been reported. In addition, age-related pharmacodynamics might also explain the variability in effect. To assess the influence of these processes on morphine effectiveness, a multi-compartment brain physiologically based pharmacokinetic/pharmacodynamic (PB-PK/PD) model was developed in R (Version 3.6.2). Active Pgp-mediated morphine transport was measured in MDCKII-Pgp cells grown on transwell filters and translated by an *in vitro-in vivo* extrapolation approach, which included developmental Pgp expression. Passive BBB permeability of morphine and its active metabolite morphine-6-glucuronide (M6G) and their pharmacodynamic parameters were derived from experiments reported in literature. Model simulations after single dose morphine were compared with measured and published concentrations of morphine and M6G in plasma, brain extracellular fluid (ECF) and cerebrospinal fluid (CSF), as well as published drug responses in children (1 day– 16 years) and adults. Visual predictive checks indicated acceptable overlays between simulated and measured morphine and M6G concentration-time profiles and prediction errors were between 1 and -1. Incorporation of active Pgp-mediated BBB transport into the PB-PK/PD model resulted in a 1.3-fold reduced brain exposure in adults, indicating only a modest contribution on brain disposition. Analgesic effect-time profiles could be described reasonably well for older children and adults, but were largely underpredicted for neonates. In summary, an age-appropriate morphine PB-PK/PD model was developed for the prediction of brain pharmacokinetics and analgesic effects. In the neonatal population, pharmacodynamic characteristics, but not

**Data Availability Statement:** All relevant data are within the manuscript and its Supporting Information files.

**Funding:** The authors received no specific funding for this work.

**Competing interests:** The authors have declared that no competing interests exist.

brain drug disposition, appear to be altered compared to adults and older children, which may explain the reported differences in analgesic effect.

## Author summary

Developmental processes in children can affect pharmacokinetics: "what the body does to the drug" as well as pharmacodynamics: "what the drug does to the body". A typical example is morphine, of which the analgesic response is variable and particularly neonates suffer more often from respiratory depression, even when receiving doses corrected for differences in elimination. One way to mathematically incorporate developmental processes is by employing physiologically based pharmacokinetic/pharmacodynamic (PB-PK/PD) models, where physiological differences between individuals are incorporated. In this study, we developed a morphine PB-PK/PD model to predict brain drug disposition as well as analgesic response in adults and children, as both processes could potentially contribute to developmental variability in the effect of morphine. We found that age-related variation in BBB expression of the main morphine efflux transporter P-glycoprotein was not responsible for differences in brain exposure. In contrast, pharmacodynamic modelling suggested an increased sensitivity to morphine in neonates.

## Introduction

Morphine is a widely used analgesic in the management of pain across the complete age range from neonates to adults. Differences in plasma morphine exposure in children result from maturation in hepatic UGT2B7 and OCT1 activity, which makes dose adjustments necessary [1,2]. Multiple population pharmacokinetic (popPK) and physiologically based pharmacokinetic (PBPK) models have been developed to account for maturational changes in morphine clearance [3–5]. In addition, developmental differences in the formation of the major active metabolite morphine-6-glucuronide and inactive morphine-3-glucuronide have been evaluated [2]. However, even with doses resulting in equivalent plasma concentrations, differences remain in the effects of morphine as indicated by frequent cases of respiratory depression in neonates and decreased morphine requirements for pain management [6,7]. Currently, it is not known whether this results from age-related changes in brain tissue pharmacokinetics or drug pharmacodynamics.

The effect of a drug relates to its concentration at the target site. Disposition into organs can be hampered particularly for more hydrophilic, permeability-limited compounds, or drugs subject to transporter-mediated transfer [8]. This is especially the case in the brain, where the blood-brain barrier (BBB) consists of endothelial cells having well-expressed tight junctions and drug efflux transporters belonging to the ATP Binding Cassette (ABC) super family. Morphine is a substrate of P-glycoprotein (Pgp)/ABCB1, which is highly expressed at the luminal side of the BBB. Differences in brain morphine disposition may explain the variability in efficacy as we and others found that BBB Pgp expression is immature in neonates and young infants [9–12]. Animal studies also point towards a developmental increase in brain Pgp activity and indicate that morphine brain concentrations increase by a factor of 1.7 in adult Pgp knocked out mice [13,14]. These findings suggest that developmental alterations in BBB Pgp expression are of clinical relevance.

An attractive alternative to animal studies is to evaluate the influence of human Pgp by *in vitro* to *in vivo* extrapolation (IVIVE) [15]. In this approach the transporter activity of Pgp

measured in an *in vitro* cell system is corrected for the amount of protein expressed and scaled with the protein abundance of the transporter in the tissue of interest. Proteomics-based determination of transporter abundance in various tissues, including the BBB, has enabled appropriate scaling [16,17]. Similarly, IVIVE has been applied successfully to abundance measurements and scaling of drug transporters in the developing liver and kidney [4,18]. In this way, developmental changes in transporter protein expression and activity can be studied, which is particularly relevant for Pgp, because there are pronounced differences in transporter activity between the human isoform and its rodent orthologs [13].

PBPK modelling is especially suited to allow pediatric predictions on tissue drug concentrations and transporter-mediated transfer in combination with IVIVE [19]. PBPK models combine knowledge on physiological processes and drug-specific properties in a multi-compartmental structure, which make them less dependent on clinically measured values compared to popPK models. This is of benefit, as plasma and tissue drug concentrations are often less widely available in a pediatric population. However, clinical data are still needed for model verification or optimization.

Apart from BBB Pgp transporter expression differences, another source of variability in morphine efficacy could originate from differences in pharmacological response between adults and children. This can be described by pharmacokinetic/pharmacodynamic (PK/PD) modeling, in which tissue-specific exposure is coupled to the receptor-mediated effect of morphine. Previously, empiric PK/PD models have been reported for adults and children, but they give limited information on the mechanisms behind the differences in PD parameters and are therefore less suitable for extrapolation to children in different age groups [1,20–22]. Better mechanistically informed models are required that include knowledge about ontogeny of relevant physiological parameters and their influence on morphine effectivity in children.

We hypothesize that age-releated differences in morphine effect can be explained by the ontogeny in Pgp expression and/or differences in pharmacodynamic response upon mu opioid receptor (MOR) binding. Here, we developed a PB-PK/PD model that incorporated both age-related Pgp activity and a PD model describing analgesic response. The PD model was based on the mutual binding of morphine and morphine-6-glucuronide to the MOR, as well as the binding-effect relationship described previously [23,24]. This model was used to investigate whether Pgp activity or PD response can be considered more important to explain the variable response in children.

## Results

### *In vitro* Pgp-mediated transport

Transport of morphine and the Pgp probe substrate digoxin (positive control) across the MDCKII-Pgp cells resulted in a net efflux ratio (ER) of 1.3 and 3.1, respectively, pointing to asymetric transport. Passive permeability of morphine (Papp,AB,inhibited) was $2.12^*10^{-6}$ cm/s (Table 1).

### Verification of pharmacokinetic profiles in adults and children

Model simulations were performed using the reported parameters in Table 1, except for morphine clearance, which was adjusted to obtain an overlay between measured and simulated plasma concentrations in adults, children and infants, to optimally assess compound disposition in brain compartments. Simulations reasonably captured measured morphine and M6G concentrations in plasma and CSF for adults and children (Fig 1A and 1B). Mean prediction errors for morphine and morphine-6-glucuronide in plasma and CSF were all between -1 and 1 (Fig 1A and 1B). Due to the moderate sensitivity of model output to the range in reported *in vitro* MDCK-Pgp expression values, this was maintained at 0.19 pmol/mg (see methods section and S1 Fig).

**Table 1. Drug-specific parameters for morphine and morphine-6-glucuronide.**

| Parameter | Description | Morphine | Morphine-6-glucuronide | Units | Notes | Ref. |
|---|---|---|---|---|---|---|
| MW | Molecular Weight | 285.343 | 461.467 | - | | [73,74] |
| LogP | Octanol-water partition coefficient | 0.89 | -2.9 | - | | [73,74] |
| pKa | Acid dissociation constant | 8.21 (base) | 2.87(acid)/ 9.12 (base) | - | | [73,75] |
| EP | Erythrocyte-plasma ratio | 1.34 | 0.15 | - | | [76] |
| PSb | Passive permeability surface area product (BBB) | $0.2112^*(Vbrain^*1.04)$ | $0.0072^*(Vbrain^*1.04)$ | L/h | | [77] |
| PSc | Passive permeability surface area product (BCSFB) | $0.2112^*(Vbrain^*1.04)^*0.5$ | $0.0072^*(Vbrain^*1.04)^*0.5$ | L/h | Surface area ~50% of BBB | [36] |
| PSe | Passive permeability surface area product (Brain-CSF) | 300 | 300 | L/h | Assumed to be no barrier | [36] |
| Fubm | Fraction unbound brain parenchyma | 0.5 | 0.99 | - | Good interspecies correlation | [61,78] |
| Fupl | Fraction unbound plasma | 0.64 | 0.83 | - | | [79,80] |
| FuCSF | Fraction unbound CSF | 1 | 1 | - | CSF protein amount low | [36] |
| Fractional morphine-6-glucuronide formation | Fraction of total morphine clearance | 0.1 (2 years–adult) 0.044 (neonates) | NA | | | [2,25,81] |
| CLiv (adult) | Intravenous clearance | $1.962^*BW$ | $0.131^*BW$ | L/h | | [82] |
| CLiv (2-18y) | Intravenous clearance | $60 * 1.62 * \left(\frac{weight}{70}\right)\left(1.47 - \frac{0.59*weight^{4.62}}{4.01^{4.62}+weight^{4.62}}\right)$ | $0.114^*BW$ | L/h | | [3,83] |
| CLiv (neonates) | Intravenous clearance | $60 * 1.62 * \left(\frac{weight}{70}\right)\left(1.47 - \frac{0.59*weight^{4.62}}{4.01^{4.62}+weight^{4.62}}\right)$ | $0.017^*BW$ | L/h | | [2,3] |
| Net *in vitro* ER | Efflux ratio in MDCKII-Pgp transwell system | 1.30 | ND | - | | *In vitro* exp. |
| Papp,AB (inhibited) | *In vitro* apparent permeability | $2.12^*10^{-6}$ | ND | cm/s | | *In vitro* exp. |

NA = Not applicable, ND = Not determined, BBB = Blood-brain barrier, CSF = Cerebrospinal fluid

ECF values were available for three adult and three pediatric patients (3.5–9.5 years). Simulated data matched with measured data, although for the 9.5 years old pediatric patient, measured data indicated that >20 hours were needed to reach equilibrium in ECF, which was underestimated in the simulations (Fig 2A). In the neonatal population, mean observed plasma concentrations were slightly underpredicted when clearance was not optimized (Fig 2B). For a separate group of neonates and young children who underwent CSF sampling (no paired plasma samples available) in which dosing regimens were variable, individual simulations were performed and compared with measured CSF data. Our model slightly underpredicted measured CSF concentrations with a mean prediction error (PE) of 0.41 (Fig 2C). Individual simulations in CSF are shown in S3 Fig. Age-related Pgp activity was included in all predictions. To evaluate the effects of Pgp activity on brain morphine ECF concentrations, simulations were performed with and without Pgp in the adult model. The presence of Pgp had a modest effect on the calculated AUC ratio (AUCno pgp/AUCpgp = 1.3) (Fig 2D).

## Verification of PD profiles in adults and children

The analgesic effect, determined as morphine-mediated increase in tolerated electrical current, was normalized to the maximum effect observed in the adult studies. No variability was included in the PD model parameters, therefore effect-time curves represent variability in

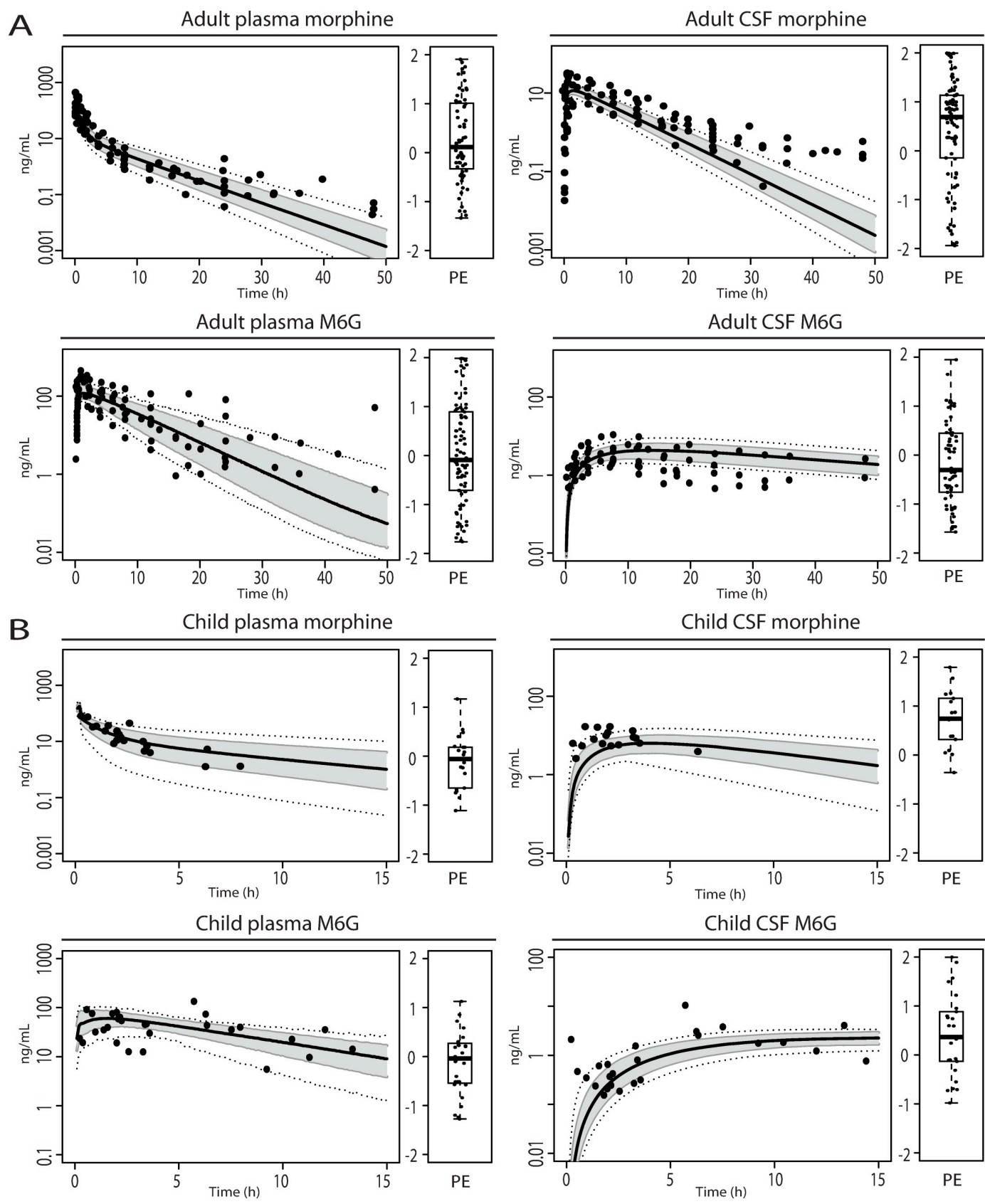

**Fig 1. Model verification of pharmacokinetic profiles in plasma and CSF of adults and children.** Panel (A): Adult morphine and M6G pharmacokinetic profiles and prediction errors (see methods, $PE = \frac{Y_{obs,i} - Y_{pred,median,i}}{(Y_{obs,i} + Y_{pred,median,i})/2}$) in plasma and CSF after single IV dose (0.38 mg/kg) morphine in neurosurgery patients. The black solid line indicates the median simulated value. The grey area represents 90% CI in inter-individual variability. Dotted lines indicate minimum and maximum simulated values. Dots are individual observed values from Meineke et al. (2002) [25]. Panel (B): Pediatric (1.4–15.9y) morphine and M6G pharmacokinetic profiles and prediction errors in plasma and CSF after single IV dose (0.25 mg/kg) morphine. The black solid line indicates the median simulated value. The grey area represents 90% CI in inter-individual variability. Dotted lines indicate minimum and maximum simulated values. Dots are observed values from Hain et al. (1999) [26].

pharmacokinetics only. Observed trends corresponded with simulated analgesic effect-time profiles (Fig 3A). The contribution of morphine and morphine-6-glucuronide to the overall effect was investigated. After a single dose of morphine, the parent compound appeared largely responsible for the drug effect, with only a minor contribution of morphine-6-glucuronide (Fig 3B). Effect-time curves were also simulated for children 2.6–16.4 years of age and neonates 10–13 days after birth. Simulated pain score profiles matched with observations for the older children, however, largely overestimated the observed pain scores for neonates (Fig 3C). The underestimation in CSF values (Fig 2C) was not the cause of the overestimated pain scores in neonates (Fig 3C), which was investigated using a simulation with 50% of the original neonatal clearance (S4 Fig).

To quantify the sensitivity of the model to: (1) *in vitro* Pgp transporter expression and (2) the morphine equilibrium dissociation constant, sensitivity analyses were performed (S1 and S2 Figs). These parameters were evaluated over the wide range of measured values that have been reported in literature (see methods section). The model appeared only moderately sensitive to variation in *in vitro* Pgp transporter expression values, while being sensitive to variability in the morphine equilibrium dissociation constant.

## Discussion

A PB-PK/PD model was developed for morphine and its active metabolite morphine-6-glucuronide. Measured data corresponded well with simulated values in brain ECF and CSF. In addition, average analgesic effect-time relationships could be predicted in adults and older children. The effect of age-related Pgp activity on morphine brain disposition appeared to be modest, and therefore seems an unlikely source of variability in morphine response in children.

The model developed in this study uses the structure proposed by Gaohua et al., which was previously extended by us to incorporate a brain barrier function based on age-related physiological parameters [36,37]. Other models including brain compartments were built on rat data and were extrapolated to human [38,39]. Previous human PBPK models predicted brain concentrations mainly for drugs passing the BBB via passive diffusion, or active transport was translated from animal species by allometric scaling [36,37,39,40]. Here, IVIVE combined with PBPK modelling proved useful for predictions of morphine brain concentrations. The same approach might be valuable for other transporter substrates in cases where tissue PK is expected to differ from plasma PK, especially when sparse clinical data is available to confirm model predicted values.

Model simulations showed that the contribution of Pgp to morphine transport across the BBB is minor, indicating that the transporter hardly influences morphine tissue concentrations in adults and children. This is in contrast to what has been described in knockout mouse models, which can be explained by lower BBB expression of Pgp in humans compared to mice [13,16]. In addition, *in vitro* studies have suggested species differences in morphine Pgp activity [13], which indicates that a quantitative animal-to-human translation is difficult. Our data is therefore at variance with the hypothesis that lower Pgp expression would be an explanation

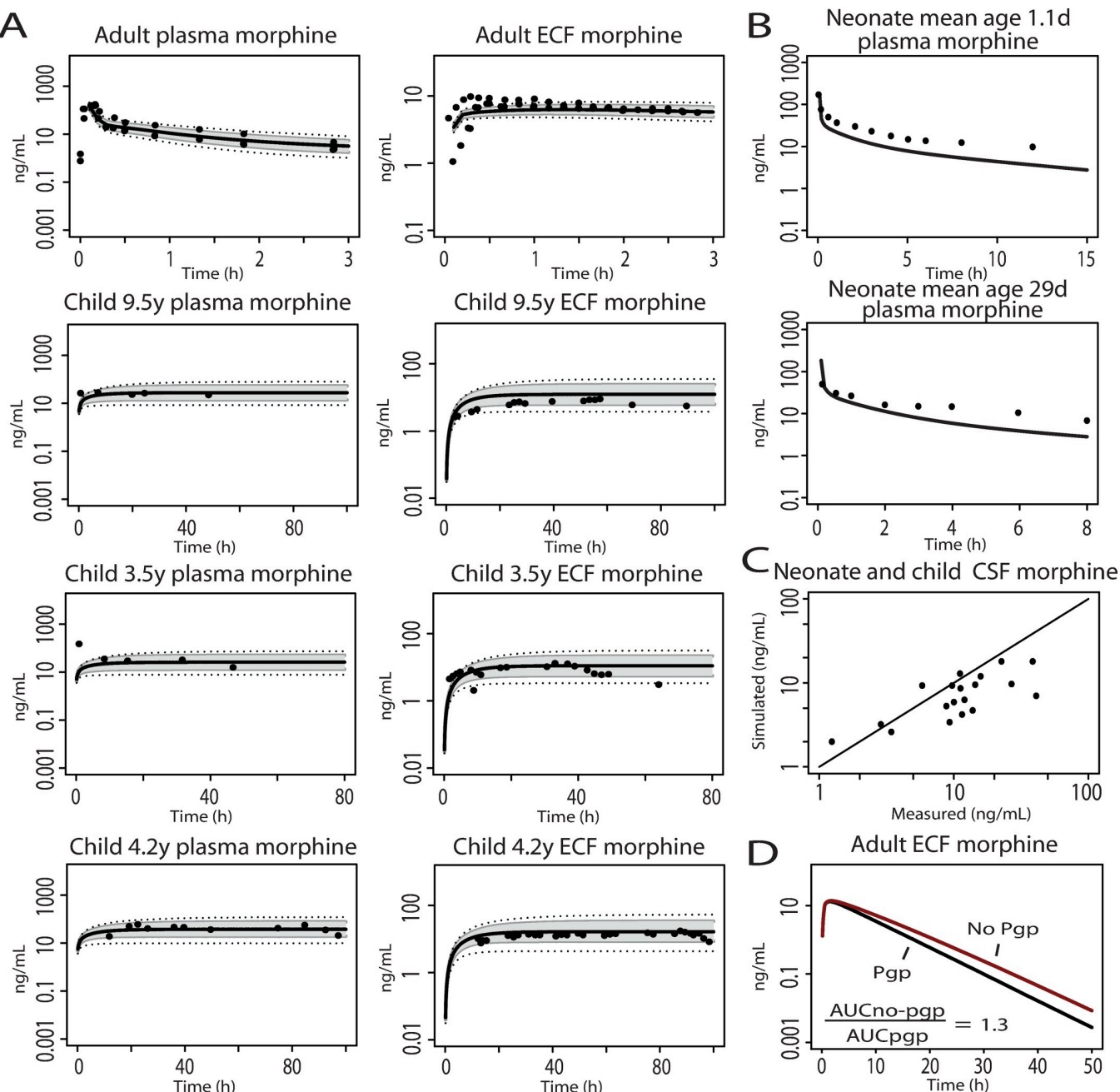

**Fig 2. Model verification of pharmacokinetic profiles in plasma, ECF and CSF of adults, children and neonates.** Panel (A): Morphine pharmacokinetic profiles in plasma and ECF after single IV infusion in adult (10 mg) and continuous IV infusion in pediatric (3.5–9.5 years, 30 μg/kg/h) traumatic brain injury patients. The black solid lines indicate median simulated values. The grey area represents 90% CI in inter-individual variability. Dotted lines indicate minimum and maximum simulated values. Dots are individual observed values from Bouw et al. (2001), Ederoth et al. (2003) and Ketharanathan et al. (2019) [27–29]. Panel (B): Predicted and observed plasma morphine concentration in neonates after a IV dose of 0.1 mg/kg (upper panel: 1.1 days mean postnatal age, lower panel: 29 days mean postnatal age). The black solid lines indicated median simulated values. Dots are mean observed values from Pokela et al. (1993) [30]. Panel (C): Median predicted v.s. measured neonatal morphine CSF values after individualized (age, dose) simulations. Observed data was derived from Radboudumc CSF biobank. Panel (D): Morphine pharmacokinetic profiles in extracellular fluid after a single dose of 0.38 mg/kg IV morphine in adult individuals. Black line: Pgp not included. Red line: Pgp included.

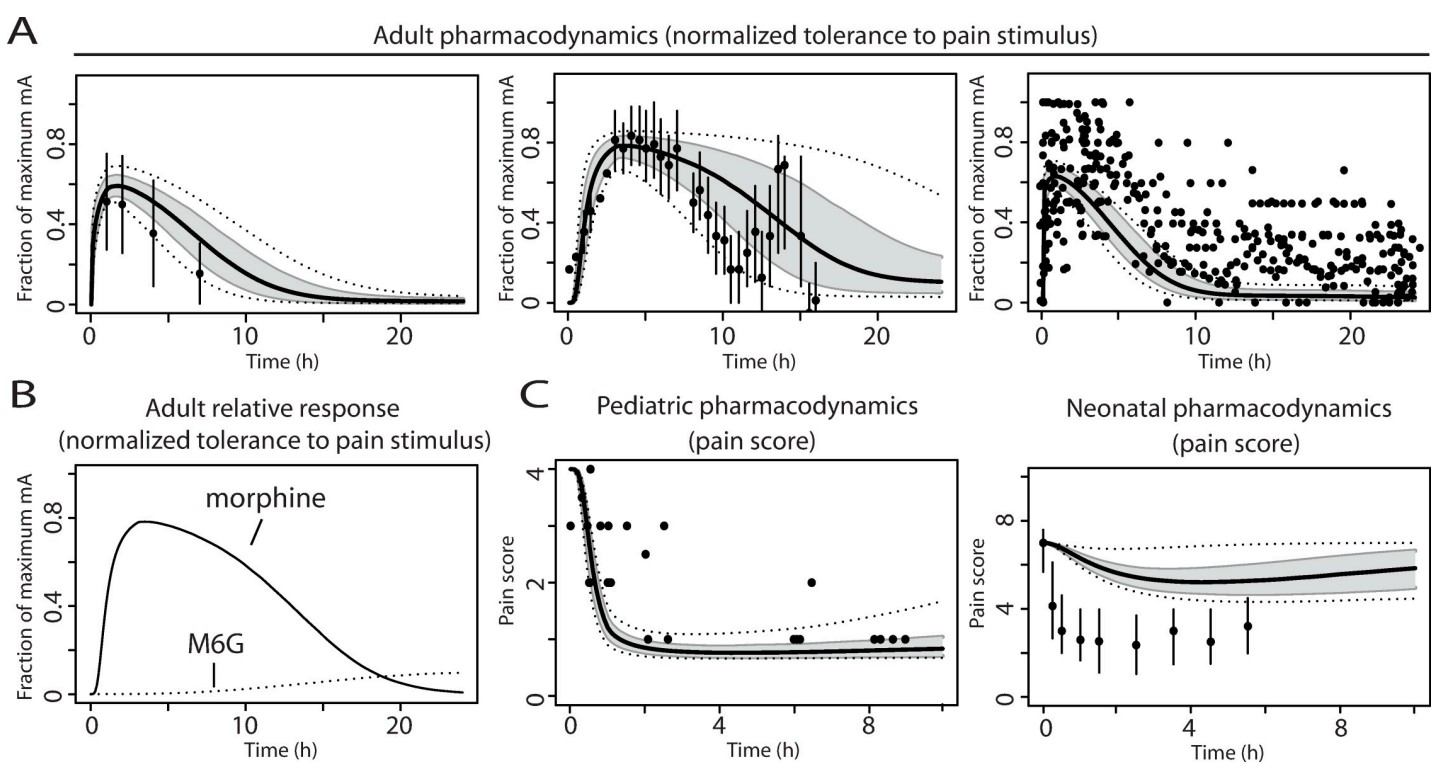

**Fig 3. Model verification of analgesic pharmacodynamic profiles in adults (pain tolerance) and children (pain score).** Panel (A): pain tolerance to an electrical pain stimulus (fraction of maximum tolerated current in milliampere) in adult healthy volunteers after (median) morphine doses from left to right of 0.13 mg/kg, 28 mg and 0.2 mg/kg IV, respectively. The black lines indicate median simulated values. The grey area represents 90% CI in inter-individual variability. Dotted lines indicate minimum and maximum simulated values. Dots represent observed values from Skarke et al. (2003), Dahan et al. (2004) and Sarton et al. (2000) [31–33]. Vertical lines indicate mean +/- S.D. values (Sarton et al. and Skarke et al. [31,33]). Panel (B): Contribution to pharmacodynamic response by morphine or morphine-6-glucuronide (M6G) over time after 28 mg morphine IV. Panel (C): pharmacodynamic pain score profiles in children after IV morphine doses of 0.25 mg/kg (children 2.6–16.4 years) and 0.03 mg/kg (neonates 10–13 days), respectively. The black lines indicate median simulated values. The grey area represents 90% CI in inter-individual variability. Dotted lines indicate minimum and maximum simulated values. Dots are observed values from Mashayekhi et al. (2009) and Enders et al. (2008) [34,35]. Vertical lines indicate interquartile range (Enders et al. [35]).

for the increased sensitivity of neonates to morphine induced respiratory depression, which was proposed previously [7].

Based on *in vitro* and animal parameters a PD model was developed, which was confirmed using human literature data. In the current model it appears that the PK-PD relationship is different in neonates compared to older children and adults, as the effect size for neonates reported by Enders *et al.*[35] was largely underpredicted. In contrast, the pain score-time profiles matched with observations in older individuals using the same binding-effect relationship for morphine [35]. Neonates therefore appear more sensitive to morphine compared to older age groups, although more clinical PD data is required to confirm this finding [19]. Especially, because the data reported by Enders *et al.* are from an open label study, which might explain the more profound decrease in measured pain score as well. In line with increased neonatal sensitivity observed here, increased sensitivity to MOR agonists has been described in rats of the same age group. Spinal injection of [D-Ala2, N-Me-Phe4, Gly5-ol] enkephalin (DAMGO) in 10 days postnatal rats, or epidural injection of morphine in 3 days postnatal rats, resulted in more pronounced analgesia compared to effects seen in older pediatric and adult rats [41,42]. This could be explained by higher neonatal MOR expression in large diameter dorsal root ganglia and dorsal horn, which are important in relaying pain signals to the central nervous system [43–45]. In addition, descending pain pathways do not appear to be mature at birth [41,46].

More elaborated PD models can be considered in the future, accounting for age-related differences in MOR expression and morphine binding-effect relationship [47]. In addition, genetic variants of the mu opioid receptor (MOR) and downstream proteins (e.g. cathecol-O-methyltransferase) can be considered when the effect on expression, binding affinity and/or the downstream cellular responses is quantified [48,49]. By combining PK and PD, this allows for more individualized dose recommendations, which currently is lacking.

This study focused on the short-term analgesic effects after morphine administration, in which morphine alone is mainly responsible for the analgesic effects. Accumulation of morphine-6-glucuronide concentrations could become significant after high and prolonged dosing, especially in cases of renal failure [50,51]. Also, morphine-3-glucuronide, currently not considered in the model due to its low affinity to MOR (Ki in micromolar range), could in this case become relevant as an antagonist [52]. Furthermore, long-term mechanisms like receptor desensitization and morphine tolerance should be accounted for in the PD model [53,54].

The model developed in this study has some important limitations and assumptions. First, no matching plasma morphine or M6G concentrations were available for the neonatal population, which would be valuable to assess the reason for the slight underprediction of measured CSF values. Second, variability was not included in the PD parameters, because the source of variation could not be clearly identified. Values on *in vitro* affinity for the MOR ranged from 1.2 to 96 nM for morphine and 0.6 to 393 nM for morphine-6-glucuronide [23]. We consider it unlikely that a >80 fold difference in affinity can be solely attributed to variation in MOR binding and assume an important influence of the experimental setup. This is an important limitation, as model outputs proved sensitive to this parameter, which would hamper prospective model development if no clinical PD data is available to select the optimal value. Future studies should identify and aim to reduce sources of variability *in vitro*, in order to allow robust incorporation into the modelling [55,56]. This could for instance be achieved by using more relevant human *in vitro* systems and accounting for receptor polymorphisms. In addition, age-related changes in binding affinity have been described in mice, which indicates that also for humans more precise estimates for different age groups could be taken into account [47].

Furthermore, the morphine binding-effect relationship was derived from a rat cortex *in vivo* study, which was based on relatively few data points [24]. From this study it appears that the receptor reserve of MOR in brain is limited, as about 80% receptor occupancy with morphine was needed for 90% of the observed effect [24]. This is consistent with a relatively low receptor reserve observed in rat after spinal morphine administration, where 55% binding resulted in 90% of the maximum effect [57]. In contrast, for (D-Ala2, N-Me-Phe4, Gly5-ol) enkephalin, a classic full MOR agonist with high efficacy, 1% binding is sufficient to achieve a similar response [57]. Morphine is found to be a partial agonist/antagonist in tissues with a low MOR expression, indicating that a relatively high receptor occupancy is required to achieve an analgesic effect [58]. In this study, the reported binding-effect relationship resulted in predictions matching human PD studies to a large extent, which indicates its suitabilty for the clinical situation. However, it is unknown how this relationship will be affected by co-medication, prolonged morphine exposure, or disease, as this can influence receptor expression. Pharmacodynamic studies therefore deserve specific attention in different sub-populations.

In conclusion, we developed a pediatric brain PB-PK/PD model for predicting morphine and morphine-6-glucuronide brain concentrations. Inclusion of BBB Pgp activity in the model had a modest effect on brain drug concentrations in adults and children and therefore appears to be not a critical factor responsible for age-related difference in morphine analgesic response. In addition, mean drug effects could be realistically predicted for adults and older children, however, were largely underpredicted for neonates, which points to an age-related difference in PD for this pediatric population.

## Methods

### Ethics statement

The Central Committee on Research Involving Human Subjects at Radboudumc waived the need for formal ethics approval according to the Dutch Law on Medical Research in Humans, as only left-over samples and electronic health record data were used.

### *In vitro* studies

*In vitro* transwell studies were performed to quantify transcellular morphine transport. Transwells (0.4 μm pore polyester membrane, Corning, Tewksbury, MA, USA) were seeded with MDCKII-Pgp cells (3*10^4 cells at 9*10^4 cells/cm2, passage number 15–23), which were kindly provided by the Netherlands Cancer Institute and cultured during 4 days with DMEM GlutaMAX (Thermo Fisher, Waltham, MA, USA), supplemented with 10% fetal calf serum. Experiments were performed on day 4 in Hank's balanced salt solution supplemented with 10mM HEPES at pH 7.4. [3H]-Morphine (American Radiolabelled Chemicals, Saint Louis, MO, USA) and [3H]-digoxin (American Radiolabelled Chemicals, positive control [59]) were added to the transwell system. To determine the A->B permeability, 100uL 0.38 uM [3H]-morphine or 76 nM [3H]-digoxin was added to the apical compartment. 600uL HBSS-HEPES buffer was added to the basolateral compartment. To determine B->A permeability, 600uL [3H]-morphine or [3H]-digoxin were added in the same concentrations to the basolateral compartment. In this case, 100uL HBSS-HEPES was added to the apical compartment. To confirm the contribution of Pgp, the same experimental setup was used in the presence of 10uM PSC833 (Tocris Bioscience, Bristol, UK), a specific inhibitor of Pgp [60], which was added to the apical compartments in all conditions. After 60 min at 37°C in an orbital shaker at 120 rpm, samples were taken from donor and receiver chambers and measured using a Hidex 35 automatic TDCR liquid scintillation counter.

Apparent permeability (Papp, cm/s) was calculated using the following equation:

$$Papp = \frac{Vr}{(S * C0)} * \left(\frac{dCr}{dt}\right) \tag{1}$$

Where Vr is the volume of the receiver compartment (mL), S is the surface area of the transwell insert (0.33 cm$^2$), C0 is the initial concentration in the donor compartment, and dCr/dt is the measured rate of change in concentration in the receiver compartment (over 60 minutes). The efflux ratio (ER) with and without the Pgp inhibitor PSC833 was derived from the ratio of the Papp in B->A direction over the Papp in the A->B direction. The net ER for Pgp was calculated by dividing the ERs of the uninhibited by the inhibited condition. Efflux ratios were used to estimate morphine Pgp transport (see below). Recovery was assessed by comparing the total amount of morphine or digoxin determined in the apical and basolateral compartments with the amount of compound added to the donor compartment, and was considered acceptable if >80%.

### Development of a PBPK model for morphine and morphine-6-glucuronide

The model was developed in R Version 3.6.2 with packages "deSolve" and "MASS" installed and consists of 14 compartments representing major organs and tissues. The 4brain model structure of Simcyp was used, which we previously modified into a pediatric version by changing physiological parameters and after verification with different drugs [36,37]. The pediatric morphine model code is provided in S1 File, and can be adjusted to an adult version by using the physiological parameters published previously [37].

In short, in the previously developed model the physiological parameters height, weight, body surface area, organ volumes, tissue flows, hematocrit, and albumin concentrations were scaled for age and/or weight [37,61]. In addition, organ partition coefficients were derived using methods described by Rodgers *et al.*, taking into account age-appropriate tissue composition, lipophilicity and ionization of compounds [62,63]. This method resulted in adult morphine and M6G Vss values of 330L and 30L, respectively. M6G Kp values were optimized using a kp-scalar of 0.5 to better reflect previously published values [64]. For the four brain compartments the following assumptions were made [37]: (1) the BBB is a barrier between blood and brain mass and the blood-cerebrospinal fluid barrier (BCSFB) separates blood and cerebrospinal fluid (CSF). No barrier exists between brain mass and CSF. (2) The BCSFB is assumed to be half of the BBB surface area, as described previously [36,39]. (3) Compartments are treated as well stirred. (4) Brain volume, brain blood flow, spinal CSF volume and CSF production rate follow age-related maturation patterns [9,61,65–68]. (5) Cranial CSF volume is described not to increase after birth [61,69]. (6) BBB surface area per gram of brain is constant, therefore total BBB is smaller in children due to a lower brain mass [70].

In this study, Pgp expression in pre-term neonates is assumed to be at 34% and at term 41% of adult expression and fully matured at 6 months of age according to immunohistochemistry data in postmortem human BBB tissue [9]. Parameters specific for neonates were extracted from the population reported in Simcyp V19 and CSF production rate in neonates was assumed to be 10 mL/h [67,68,71,72]. Drug-specific model parameters for morphine and morphine-6-glucuronide are listed in table 1 and were derived from literature sources or *in vitro* data generated in this study.

Total body clearance values were based on (pediatric) population PK studies. Passive brain morphine and morphine-6-glucuronide clearance permeability data were obtained from a rat *in vivo* perfusion study [77]. Fraction of unbound morphine in brain was derived from primate data because of the good correlation with human data [78,84]. The fraction of unbound morphine-6-glucuronide was estimated using the Simcyp Fubm predictor, as no experimental value is available from literature [61]. Active Pgp-mediated morphine transport across the BBB was calculated from *in vitro* data (previous section) and scaled to the *in vivo* activity parameter by the following equation described by Li *et al.* [40].

$$CLefflux, vitro = \frac{2*(ER-1)*Papp, AB(inhibited)*SA}{Procell} \qquad (2)$$

Where ER is the *in vitro* morphine net efflux ratio, Papp, AB(inhibited) is the apparent permeability in the MDCKII-Pgp cells in apical to basolateral direction when Pgp is inhibited ($2.12^*10^{-6}$ cm/s), SA is the surface area of the transwell filter (0.33 cm$^2$) and Procell is the amount of cellular protein measured on a transwell filter (81,4 μg).

CLefflux,vitro was scaled to whole BBB efflux clearance (CLefflux,vivo) by Eq 3.

$$CLefflux, vivo = CLefflux, vitro * \frac{abundance(ex\ vivo)}{abundance(in\ vitro)} * BMvPGB * BW \qquad (3)$$

Where CLefflux,vitro is the *in vitro* clearance calculated using Eq 2, abundance (*ex vivo*) is the Pgp protein abundance in adult isolated microvessels, which was reported to be 4.21 pmol/mg total protein on average [16,17,85]. Pgp abundances (*in vitro)* in MDCKII-Pgp cells reported in literature range from 0.19 to 7.02 pmol transporter/mg total protein [40,86,87]. Pgp abundance reported in pmol transporter/mg membrane protein was converted to pmol transporter/mg total protein by the ratio reported by Ohtsuki *et al.* [87]. Initially, 0.19 pmol transporter/mg total protein was used as this results in maximum *in vivo* efflux clearance

(CLefflux,vivo). BMvPGB is the amount of brain microvessel protein per gram brain tissue (0.244 mg/g) and BW is the brain weight (1400 g) [40].

## PD model development

With the PBPK model introduced in the previous section, unbound brain concentrations were calculated that can be used to predict the amount of receptor-bound morphine and morphine-6-glucuronide. The fraction of bound receptor was calculated using Eqs 4 and 5.

$$\%BRM = \frac{[M]}{[M] + KM\left(1 + \frac{[M6G]}{KM6G}\right)} * 100 \tag{4}$$

$$\%BRM6G = \frac{[M6G]}{[M6G] + KM6G\left(1 + \frac{[M]}{KM}\right)} * 100 \tag{5}$$

Where %BRM and %BRM6G represent the percentages of MOR bound by morphine and morphine-6-glucuronide, respectively; [M] and [M6G] are the unbound brain concentrations of morphine and morphine-6-glucuronide; KM and KM6G are the equilibrium dissociation constants for morphine and morphine-6-glucuronide. Equilibrium dissociation constants for MOR have been reviewed previously and average values of 11.8 nM for morphine and 42.5 nM for M6G were used, as this resulted in the best fit of the model to measured adult PD data [23]. The equations describing competitive binding resulted in a maximum fraction of bound receptor (by morphine plus morphine-6-glucuronide) of 1 and allowed calculation of the fraction of receptors that are bound by morphine and morphine-6-glucuronide.

A relationship between the percentage of morphine bound receptor and relative analgesic effect was established in a rat study [24]. Here, we used nonlinear regression analysis (Graphpad Prism version 5.03) to fit the following sigmoid exposure-response equation to the data:

$$RR = 100/(1 + (BR50/[BR]) * Hill\ Slope) \tag{6}$$

Where RR is the relative response (in %), BR50 is the percentage bound receptor resulting in 50% effect, which was estimated at 59.3 (95% CI 54.1–64.4). BR is the percentage of MOR bound by morphine. The Hill slope was estimated at 4.2 (95% CI 2.6–5.8).

The parameters for morphine-6-glucuronide in Eq 6 were optimized in order to obtain a potency difference of 2 compared with morphine (BR50: 17, Hill slope: 2.2), as a difference of 2 and 2.6 between potencies has been reported after intravenous and intrathecal administration [88,89]. Concentration-effect curves combining Eqs 4 and 6, as well as 5 and 6, can be found in S5 Fig. The relative effects calculated were multiplied with the maximum observed effect in the clinical studies used (see below). Due to the unknown sources of variability in the effect parameters, this was currently not considered in the PD model.

## Clinical data for model validation

Published clinical data describing plasma, CSF and extracellular fluid (ECF) concentration-time profiles or analgesic effect-time profiles of morphine and morphine-6-glucuronide were used for model verification and, if necessary, extracted from the original publication using WebPlotDigitizer version 4.1 (Table 2).

**Morphine and morphine-6-glucuronide plasma, CSF and ECF measurements in adults and children.** All samples were derived from patients with underlying disease, as the samples were taken in the context of clinical care (Table 2) [25–29]. For adults, a study was used where plasma was obtained and CSF was sampled from a ventricular catheter in neurosurgical

patients (19–69 years) [25]. To validate the model in children, data was extracted from a study in pediatric leukemia patients (1–18 years) who underwent diagnostic lumbar puncture to obtain CSF and paired blood sampling [26]. In both studies morphine and morphine-6-glucuronide concentrations were quantified. In addition, morphine plasma and ECF drug concentrations were available for three adult (32–52 years) and three pediatric (3.5–9.5 years) traumatic brain injury patients, which were sampled using a microdialysis catheter in brain parenchyma [27–29].

No data was available for young children 0–1 years of age. Therefore, we obtained 19 CSF patient samples (age range: 0–11 months postnatal age, median 3 days postnatal age), which were collected in the setting of clinical care and stored in the Radboudumc CSF Biobank, Nijmegen, Netherlands. Morphine concentrations were determined using LC-MS/MS quantification methods as described previously by de Bruijn *et al.* [90]. Age and individual dosing regimens were extracted from the electronic patient records to allow patient-specific simulations. Due to the absence of paired plasma data in our cohort, the neonatal plasma model output was compared with mean measured plasma concentrations from Pokela et al., similar to what has been described previously [30,91].

**Pharmacodynamic studies in adults and children.** Three studies (Table 2) were used for verification of the pharmacodynamic model in healthy adult volunteers (21–36 years). A transcutaneous electrical acute pain stimulus was applied to evaluate the short-term analgesic effect of morphine at regular time intervals [31–33]. Effects of morphine were reported as the increase in electrical current in milliampere tolerated (morphine minus mean placebo values, if available). Drug effects were normalized to the highest effect observed in the study to allow comparison between the adult studies. For one adult study, values were normalized to the highest effect observed for each individual patient, due to the large inter-individual variability [32]. Pediatric pain scores from children were derived from a clinical study in cancer patients (2.6–16.4 years) receiving morphine [34]. Pain was scored on a scale from 0 (no pain) to 4 (severe pain). In preterm neonates (10–13 days postnatal age, 29.8 weeks gestational age) with clinical signs of pain due to application of continuous positive airway pressure, the analgesic effect after morphine dosing was evaluated using a 10-point visual analog scale (0 means no pain, 10 severe pain) [35].

Simulations were run for 500 individuals matched with the clinical study for dosing regimen, age, percentage male/female. Simulated plasma concentrations, unbound brain concentration and CSF median, 5th percentile, 95th percentile, minimum, and maximum concentration profiles were overlaid with data derived from clinical studies described above. Due to the absence of the ECF compartment in the model, predicted unbound brain concentrations were used to simulate measured unbound ECF values. Median analgesic effect-time profiles were compared with measured data. Prediction errors were calculated for plasma and CSF drug concentrations as described previously [38], following the equation:

$$PE = \frac{Yobs, i - Ypred, median, i}{(Yobs, i + Ypred, median, i)/2} \tag{7}$$

Where Yobs,i is the ith observed concentration in the clinical studies at a specific point in time and Ypred,median,i is the ith median predicted value for the same point in time. Variability between subjects was expected to cancel out in the analysis, therefore, the average PE ideally equals 0. An average PE of +/- 0.667 and +/- 1 refer to a 2-fold or a 3-fold average difference between predicted and observed values, respectively. No PE was calculated for the comparison between observed and predicted ECF values, as measurements from only three individuals were available.

**Table 2. Characteristics of studies included for model verification.**

| | | | Pharmacokinetics | | | |
|---|---|---|---|---|---|---|
| Study | Number of patients | (median) morphine dose | Co-medication | Age (y) | Indication | PK sample collection |
| Meineke *et al.*, 2002 [25] | 9 | 0.38 mg/kg signle IV dose | acetyl-cysteine; amoxicilline; cefotaxime; cefuroxime; ceruletide; cisapride; clonidine; dexamethasone; dimeticone; dopamine; furosemide; flunitrazepam; gentamicin; insulin; ketamine; lactulose; metoclopramide; meropenem; metronidazole; midazolam; molsidomine; nystatin; nimodipine; ofloxacin; omeprazole; paracetamol; sorbide; urapidil | 19-69y | Neurological and neurosurgical patients | Plasma, CSF (ventricular drain) |
| Ederoth *et al.*, 2003 [28] | 2 | 10 mg sigle IV dose | midazolam, fentanyl | 32y, 52y | Trauma patients | Plasma, ECF (microdialysis) |
| Bouw *et al.*, 2001 [27] | 1 | 10 mg single IV dose | NA | 52y | Trauma patient | Plasma, ECF (microdialysis) |
| Hain *et al.*, 1999 [26] | 17 | 0.25 mg/kg single IV dose | NA | 1-18y | Acute leukemia | Plasma, CSF (Lumbar puncture) |
| Ketharanathan *et al.*, 2019 [29] | 3 | 0.03 mg/kg/h continuous IV dose | NA | 3.5y, 4.2y, 9.5y | Trauma patients | Plasma, ECF (microdialysis) |
| | | | Pharmacodynamics | | | |
| Study | Number of patients | (median) morphine dose | Co-medication | Age (y) | Indication | PK and PD sample collection |
| Sarton *et al.*, 2000 [33] | 20 | 0.1 mg/kg single IV dose, 0.03mg/kg/h IV dose during 1 h. | - | 21-36y | Healthy | Plasma, Tolerance to electrical current |
| Dahan *et al.*, 2004 [32] | 16 | 0.2 mg/kg single IV dose | - | 18-24y | Healthy | Plasma, Tolerance to electrical current |
| Skarke *et al.*, 2003 [31] | 12 | 28 mg single IV dose | - | 23-29y | Healthy | Plasma Tolerance to electrical current |
| Mashayekhi *et al.*, 2009 [34] | 4 | 0.25 mg/kg/h IV dose during 1 h. | NA | 2.6–16.4y | Cancer patients | Plasma, Llandough Hospital pain assessment |
| Enders *et al.*, 2008 [35] | 64 | 0.025 mg/kg single IV dose | NA | 10-13d | continuous positive airway pressure associated pain | Visual analog scale |

NA = Not available, CSF = Cerebrospinal fluid, ECF = Extracellular fluid

## Supporting information

**S1 Fig. Sensitivity analysis MDCK-Pgp transporter protein expression.** In vitro, MDCK Pgp protein abundance is reported to range from 0.19–7.02 pmol Pgp/mg total protein. Simulations were performed in adults and neonates after a dose of 28 mg or 0.03 mg/kg, respectively. Black lines indicate predictions using the default value of 0.19 pmol Pgp/mg MDCK total protein (100% Pgp activity), grey lines indicate predictions using 0.48 pmol Pgp/mg MDCK total protein (40% Pgp activity), and red lines indicate predictions where no Pgp is included (0% Pgp activity).
(EPS)

**S2 Fig. Sensitivity analysis morphine equilibrium dissociation constant.** Reported morphine equilibrium dissociation constants (Kd) range from 1.2 to 96nM in literature. Simulations were performed in adults and neonates after a dose of 28 mg or 0.03 mg/kg, respectively.

Black lines indicate predictions using the default Kd value of 14.8 nM, dark grey lines indicate predictions using a Kd value of 1.2nM and light grey indicate predictions using a Kd value of 96nM.
(EPS)

**S3 Fig. Individual neonatal and pediatric morphine CSF predictions (variable dose regimen).**
(EPS)

**S4 Fig. Model prediction of pharmacokinetic profiles in CSF of neonates and young infants at 50% of the original clearance and corresponding predictions of pain scores in neonates.** Panel (A): Predicted and observed plasma morphine concentration in neonates after an IV dose of 0.1 mg/kg (left panel: 1.1 days mean postnatal age, right panel: 29 days mean postnatal age). Black solid lines indicated simulated values. Dots are observed values from Pokela et al. (1993) [30]. Panel (B): Median predicted v.s. measured neonatal morphine CSF values after individualized (age, dose) simulations. Observed data was derived from Radboudumc CSF biobank. Panel (C): pharmacodynamic pain score profiles in neonates after an IV morphine dose of 0.03 mg/kg (neonates 10–13 days). The black lines indicate median simulated values. The grey area represents 90% CI in inter-individual variability. Dotted lines indicate minimum and maximum simulated values. Dots are observed values from Enders et al. (2008) [35]. Vertical lines indicate interquartile range.
(EPS)

**S5 Fig. Morphine and Morphine-6-glucuronide concentration-effect relationship.** Parameter values and abbreviations are described in the methods section.
(EPS)

**S1 File. R code pediatric morphine PB-PK/PD model.**
(R)

## Author Contributions

**Conceptualization:** Laurens F. M. Verscheijden, Jan B. Koenderink, Saskia N. de Wildt, Frans G. M. Russel.

**Data curation:** Laurens F. M. Verscheijden.

**Formal analysis:** Laurens F. M. Verscheijden.

**Investigation:** Laurens F. M. Verscheijden, Carlijn H. C. Litjens.

**Methodology:** Laurens F. M. Verscheijden, Carlijn H. C. Litjens, Ron H. J. Mathijssen, Marcel M. Verbeek.

**Resources:** Ron H. J. Mathijssen, Marcel M. Verbeek.

**Supervision:** Jan B. Koenderink, Saskia N. de Wildt, Frans G. M. Russel.

**Validation:** Laurens F. M. Verscheijden.

**Visualization:** Laurens F. M. Verscheijden.

**Writing – original draft:** Laurens F. M. Verscheijden, Saskia N. de Wildt, Frans G. M. Russel.

**Writing – review & editing:** Carlijn H. C. Litjens, Jan B. Koenderink, Ron H. J. Mathijssen, Marcel M. Verbeek, Saskia N. de Wildt, Frans G. M. Russel.

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
