## [Decision Letter · Decision Letter 0]

13 Nov 2020

Dear Mr. Verscheijden,

Thank you very much for submitting your manuscript "Physiologically based pharmacokinetic/pharmacodynamic model for the prediction of morphine brain disposition and analgesia in adults and children" for consideration at PLOS Computational Biology.

As with all papers reviewed by the journal, your manuscript was reviewed by members of the editorial board and by several independent reviewers. In light of the reviews (below this email), we would like to invite the resubmission of a significantly-revised version that takes into account the reviewers' comments.

We cannot make any decision about publication until we have seen the revised manuscript and your response to the reviewers' comments. Your revised manuscript is also likely to be sent to reviewers for further evaluation.

Sincerely,

James Gallo

Associate Editor

PLOS Computational Biology

Mark Alber

Deputy Editor

PLOS Computational Biology

Reviewer's Responses to Questions

**Comments to the Authors:**

Reviewer #1: Major

1) The authors should provide a model validation in neonates and infant for plasma concentrations which should be similar to the model validation for adults and children in Figure 1. Previously published concentration time profiles could be found in the following publications:

Pokela ML, Olkkola KT, Seppala T, Koivisto M. Age-related morphine kinetics in infants. Dev Pharmacol Ther. 1993;20(1-2):26–34.

Olkkola KT, Maunuksela EL, Korpela R, Rosenberg PH. Kinetics and dynamics of postoperative intravenous morphine in children. Clin Pharmacol Ther. 1988;44(2):128–136

A validated systemic concentration time profile is the prerequisite for the development of brain compartments for CSF concentrations. Without these validations, it is difficult to evaluate the reason for the biased model prediction in Figure 1D.

2) In this manuscript, pain scores were evaluated in healthy volunteers, oncology patients, and neonates. Different studies accessed different types of pain with different clinical instruments. It might not be possible or reasonable to model the different pain scores using the same PD model for different pain instruments and types of pain. Also, the authors did not discuss the pharmacodynamic development in pediatrics. If pediatrics responded to opioids differently throughout postnatal development, it may also support a different pain model for neonates (e.g. Journal of Neuroscience 1 April 1998, 18 (7) 2538-2549 and PAIN155 (2014) 168–178169). The authors should also notice that the model predicted concentrations in Figure 1D were lower than the observed concentrations, which might be another reason for the underestimated response in neonates in Figure 2.

Minor

1) Since the transporter has an impact on distribution, the authors should also demonstrate that the morphine distribution (Vss) is reasonably estimated in adults and children. Although volume of distribution is estimated based on chemical properties in a PBPK model, the authors should provide the estimation method and the estimated volume of distribution for morphine and M6G. Also, these estimated values should be close to the previously published steady state volume of distribution for morphine after IV administration of morphine (e.g. Clin Pharmacol Ther. 2002 Aug;72(2):151-62) and for M6G after IV administration of M6G (e.g. Acute Pain (2006) 8, 63—71).

2) Rstudio (Version 1.1.442) is a GUI not a software or package. In the manuscript, the authors should specify the R version and the packages that were used to develop the PBPK model.

3) It will be easier to read if the unit of ng/mL and decimal values were used on y-axis for morphine and M6G concentrations in Figure 1.

4) In Figure 2, three out of the six figures used the observed individual data to evaluate the model predictions. However, it is difficult to tell the central tendency of the individual data. Please use mean +/- standard deviation to replace the individual data points.

Reviewer #2: The present work addresses an existing knowledge gap related to the understanding of the underlying sources of morphine response variability in pediatric patients. The authors present an application of PBPK modeling utilizing IVIVE with an incorporated PD model to assess the contribution of Pgp activity and simulated MOR receptor occupancy towards age-related variable morphine response. I find that the work has merit, owing to the relatively novel application of PBPK modeling with additional mechanistic insight of morphine PD for addressing a significant issue in pediatric health care. In current drafting, the primary weaknesses of the work stem from a want for technical details related to the application of in vitro data and apparent absence of the supplemental methods section. My recommendations for major and minor corrections are detailed below:

Major comments:

1. The supplementary methods section S1 appears to be missing from the document on page 21 – there is a subsection title, but no subtext beneath it prior to the references subsection. The supplied hypertext link included on the final page of the pdf document leads to a download of the supplemental figure only, and no additional source files appear on the reviewer task page. The supplementary methods for morphine and M6G C/E relationship will need to be reviewed prior to rendering a final disposition on this manuscript.

2. A primary conclusion of the study is that Pgp is unlikely to play a significant role in age-related, variable morphine response. Notably, assumptions are made in the application of IVIVE of Pgp activity from cell culture-derived data, and detail is also missing from discussion of the in vitro methods. This makes it difficult to determine the validity of the conclusion. It is understandably out of reach for the typical wet lab to quantitate enzyme expression levels using advanced MS-based proteomics, but ideally there should be some methodology in place to ensure standardization, and if feasible, a qualitative assessment of expression levels (e.g. western blot against a known standard). The main concern here is that the literature range of 0.19 – 7.02 pmol/mg protein seems quite wide, and in the Methods section it is only discussed that 0.19 pmol/mg was initially investigated. Specific questions I have regarding this application include:

a. Was the full range of 0.19 – 7.02 pmol/mg abundance of Pgp investigated, and were model sensitivity analyses done on this parameter? If the model results were not significantly different across this range, or if values above 0.19 pmol/mg produce unreasonable model performance then it should be stated as such in the text.

b. If the model predictions are sensitive across this range, it will be useful to know if cell culture conditions were consistent between experiments and between controls of individual experiments. The most salient variable in this regard should be cell passage number after reviving cells from cryo-storage. Please add any known relevant details regarding how cell culture conditions were controlled.

c. If any of the above conditions constitutes a limitation on interpreting the Pgp contribution to model predictive performance, please include a brief statement in the Discussion section.

3. Similar to point 2 above, the data used for MOR ki values for morphine and M6G are average values from tabulated data in reference 23 (Osbourne et al., 2000). Ranges are given on Page 10, Lines 237-8: which derives from 10 prior in vitro studies using seven different tissue models and seven different radiolabeled ligands. The large variability of this data is mentioned in the Discussion section (as it should be) but the current work would benefit from additional commentary on the sensitivity of the PD model predictive performance to ki values. At the very least, I would suggest for median ki values also be investigated/discussed since the distributions for both morphine and M6G are significantly biased and the fold difference between median and average values are between ~3 - 4.

Requests for minor corrections:

1. Page 4, line 82: I suggest using a more specific term in place of “drug efficacy” here. I would think to simply say “pharmacodynamics” though I’m unsure if that captures your intent.

2. Page 4, line 90-91: I missed a reference for a developmental increase in brain Pgp activity as this does not appear to be covered by reference 13 (Schinkel et al., 1995)

3. Page 5, line 100: reference 18 as listed in the bibliography also does not appear to support the statement of successful application of IVIVE to transporter abundance/scaling as it only discusses abundance of Phase-II UGT family enzymes.

4. Page 6, Lines 134-136: (Figure 1B) The model appears somewhat biased towards underpredicting peak, child CSF concentration for morphine. While I don’t think this is a huge problem, I was curious about the correlation to the PD model performance (Figure 2C) which may appear to overpredict morphine effect. Do the authors have any thoughts?

5. Page 6, Line 135: text mentions “drug concentrations”, but should probably also list “metabolite”

6. Page 7, Line 168: No variability was included for the PD model parameters. (see major comment 3 above)

7. Page 12, Line 271: please provide also the cell-seeding density (ie 3*10^4 cells at 9.1*10^4 cells/cm^2)

8. Page 13, Line 297: Please list the R packages and any specific libraries used for PBPK model development, construction and validation.

9. Page 19, line 405: Text here lists range for pediatric/adolescent pain scores as 0 to 3, yet Figure 2C has data points at 3.5 and 4.

Reviewer #3: Given unexpected time constraints, I unfortunately did not have the time to review this manuscript with the thoroughness it would merit, and I kindly ask the editor to consider the lack of time to conduct an in-depth review in his final assessment.

The manuscript addresses an important topic, though, and appears to be adequately written.

Minor comments:

Introduction

“The effect of a drug relates more closely to its tissue rather than plasma concentration, particularly for more hydrophilic, permeability-limited compounds, or drugs subject to transporter-mediated transfer” – I believe this statement is not true; the effect of a drug rather relates to the drug concentrations at the target site. The drug target can also be in the blood (e.g. anticoagulants) or in the GI lumen (e.g. antacids, some anthelmintics). I suggest rephrasing.

“PBPK models combine knowledge on physiological processes and drug-specific properties in a multi-compartmental structure, which make them less dependent on clinically measured values compared to popPK models” This is also a very bold statement; please be more precise: PBPK predictions are less dependent on clinical data, but in many cases you need these data nonetheless to evaluate the model. Otherwise the value of the predictions cannot be assessed. I suggest rephrasing.

“Another source of variability in morphine efficacy could originate from…” What is the first source? Maybe rephrase as “Variability in morphine efficacy could originate from…”

“Currently, mechanistic PK/PD models are scarce both for adults and children” I disagree with this statement. There are certainly more mechanistic PK model published, but I don’t have the impression that mechanistic PD models are scarce.

“We hypothesize that age-related differences in morphine effect can be explained by the ontogeny

in Pgp expression and/or differences in pharmacodynamic response upon mu opioid receptor (MOR)

binding” Maybe you could elaborate a bit on the mechanism of altered pharmacodynamic response upon receptor binding. From the next sentence I get the impression that altered PD is exclusively modeled by different tissue concentrations of morphine and morphine-6-glucuronide to the MOR which ultimately would be related again to PK but not to different PD.

Figure 1: Why are the PE plots below the PK profile plots not given for all PK plots but only for panel A and B?

Figure 1: Please provide a zoom-in for some panels; the axis limits of some PK profiles are too squeezed, and the plots are therefore not very informative. It may even be better to separate Fig. 1 in multiple figures that have a larger size/resolution.

Figure 1D: Can you also provide the PK profile for the neonates? This will be more informative than the GOF plot.

Figure 1E: Please add the unit of the y-axis (mg/L ,I assume)

Can the mismatch between observed and predicted neonatal pain score (Fig. 2C) be fully explained by the relatively poor predictive performance of the neonatal CSF morphine predictions? This question could be further explored through e.g. conducting a sensitivity analysis.

I think the discussion could be improved if the technical aspects of model are more broadly imbedded in what has been published previously by other groups. E.g. how is this model structure and parameterization and the implemented ontogeny different from other models?

Since the model is developed in R will it be made available to the community upon publication? I think this would be of great value. These days I see too many models that are not available open access to the community and independent reproduction of these models often fails which I believe undermines in the long term the confidence in these approaches.

**Have all data underlying the figures and results presented in the manuscript been provided?**

Reviewer #1: Yes

Reviewer #2: Yes

Reviewer #3: None

PLOS authors have the option to publish the peer review history of their article (what does this mean?). If published, this will include your full peer review and any attached files.

Reviewer #1: No

Reviewer #2: No

Reviewer #3: No
---

## [Decision Letter · Decision Letter 1]

21 Jan 2021

Dear Mr. Verscheijden,

Thank you very much for submitting your manuscript "Physiologically based pharmacokinetic/pharmacodynamic model for the prediction of morphine brain disposition and analgesia in adults and children" for consideration at PLOS Computational Biology. As with all papers reviewed by the journal, your manuscript was reviewed by members of the editorial board and by several independent reviewers. The reviewers appreciated the attention to an important topic. Based on the reviews, we are likely to accept this manuscript for publication, providing that you modify the manuscript according to the review recommendations.

Sincerely,

James Gallo

Associate Editor

PLOS Computational Biology

Mark Alber

Deputy Editor

PLOS Computational Biology

[LINK]

Reviewer's Responses to Questions

**Comments to the Authors:**

Reviewer #1: The authors have addressed almost all the comments I had previously except for my major comment #1. The authors' simulation for neonatal CSF concentration addressed part of the concern on pharmacodynamic response. However, I still have the following comment regards to the same request I had previously:

The external evaluation for plasma concentration in neonates is not for the explanation of biased pharmacodynamic prediction but also for the evaluation of PBPK model. The authors collected clearance model from different publications (e.g., Wang et al.) for morphine and M6G and forced it into the PBPK model. The original model assumptions and misspecifications may not remain validate in the new PBPK model. In the response to my major comment 1, the authors simply claimed that they cannot be sure that the recommend external data fully reflect the population from which the authors collected in this study. However, the author did not explain what the difference between these two populations is and how it may lead to a different PK characteristic. The authors claim seems to be bold. Unless there's specific disease related factors that may change PK, I would believe PK should be close enough in different patient populations. Therefore, I still believe that the authors should provide the external evaluation of neonatal plasma concentration profile. I recommend the authors follow the external evaluation step in the following article to evaluate their PBPK model.

Liu et al. Mechanistic population pharmacokinetics of morphine in neonates with abstinence syndrome after oral administration of diluted tincture of opium. The Journal of Clinical Pharmacology 56(8).

Particularly, in this publication, it showed how one integrated morphine PK model could describe PK across different age groups as an external evaluation. The two popPK model cited (Wang and Bouwmeester) in the current manuscript are both included in this publication. Morphine population pharmacokinetics model and PBPK model have been published previously many times. I do believe we should have a better model in the following publications, and an external evaluation can tell us directly how these models work in different aspects.

Reviewer #2: Upon review, I find the authors have adequately responded to reviewer comments and suggestions and that the revised manuscript is significantly improved by their efforts. In this reviewer's opinion, one of the strengths of this work is the demonstrated use of open-source software for the development of PBPK models in "special" or "difficult-to-treat" populations.

Reviewer #3: Thank you for the careful revision

**Have all data underlying the figures and results presented in the manuscript been provided?**

Reviewer #1: Yes

Reviewer #2: Yes

Reviewer #3: None

PLOS authors have the option to publish the peer review history of their article (what does this mean?). If published, this will include your full peer review and any attached files.

Reviewer #1: No

Reviewer #2: **Yes: **David R. Hahn, PhD

Reviewer #3: No
---

## [Decision Letter · Decision Letter 2]

12 Feb 2021

Dear Mr. Verscheijden,

We are pleased to inform you that your manuscript 'Physiologically based pharmacokinetic/pharmacodynamic model for the prediction of morphine brain disposition and analgesia in adults and children' has been provisionally accepted for publication in PLOS Computational Biology.

Best regards,

James Gallo

Associate Editor

PLOS Computational Biology

Mark Alber

Deputy Editor

PLOS Computational Biology

Reviewer's Responses to Questions

**Comments to the Authors:**

Reviewer #1: The external evaluations in neonates looks good. I appreciate the authors modifications. No further comments.

**Have all data underlying the figures and results presented in the manuscript been provided?**

Reviewer #1: Yes

PLOS authors have the option to publish the peer review history of their article (what does this mean?). If published, this will include your full peer review and any attached files.

Reviewer #1: No

---

## [Editor Report · Acceptance letter]

26 Feb 2021

PCOMPBIOL-D-20-01761R2 

 Physiologically based pharmacokinetic/pharmacodynamic model for the prediction of morphine brain disposition and analgesia in adults and children 

Dear Dr Verscheijden,

I am pleased to inform you that your manuscript has been formally accepted for publication in PLOS Computational Biology. Your manuscript is now with our production department and you will be notified of the publication date in due course.

With kind regards,

Alice Ellingham
